# Utilizing Human Behavior Modeling to Manipulate Explanations in AI-Assisted Decision Making: The Good, the Bad, and the Scary

**Zhuoyan Li**
Department of Computer Science
Purdue University
West Lafayette, IN, 47906
li4178@purdue.edu

**Ming Yin**
Department of Computer Science
Purdue University
West Lafayette, IN, 47906
mingyin@purdue.edu

## Abstract

Recent advances in AI models have increased the integration of AI-based decision aids into the human decision making process. To fully unlock the potential of AI-assisted decision making, researchers have computationally modeled how humans incorporate AI recommendations into their final decisions, and utilized these models to improve human-AI team performance. Meanwhile, due to the "black-box" nature of AI models, providing AI explanations to human decision makers to help them rely on AI recommendations more appropriately has become a common practice. In this paper, we explore whether we can quantitatively model how humans integrate both AI recommendations and explanations into their decision process, and whether this quantitative understanding of human behavior from the learned model can be utilized to manipulate AI explanations, thereby nudging individuals towards making targeted decisions. Our extensive human experiments across various tasks demonstrate that human behavior can be easily influenced by these manipulated explanations towards targeted outcomes, regardless of the intent being adversarial or benign. Furthermore, individuals often fail to detect any anomalies in these explanations, despite their decisions being affected by them.

## 1 Introduction

Recent advances in AI models have significantly increased the integration of AI-based decision aids into human decision making process. The widespread adoption of such AI-based decision aids has opened up a new paradigm of human-AI collaboration–the AI model provides recommendations for a given decision making task, while human decision makers are responsible for making the final decisions. To fully unlock the potential of AI-based decision aids in enhancing human decision making, a few studies [1–5] have developed computational models to capture how humans factor AI recommendations into their decision-making process, and explored how these behavioral models can be utilized to improve human-AI team performance. For example, Vodrahalli, Gerstenberg, and Zou [6] developed a human behavior model to characterize the impact of AI predictions and confidence levels on human final decisions. This model was then utilized to adjust the model confidence displayed to people, with the objective of calibrating human trust in AI assistance.

Meanwhile, the black-box nature of prevalent AI models has driven a greater integration of model explanations, generated through various explainable AI (XAI) methods [7–11], into AI-assisted decision making. These explanations seek to provide some insights into the the underlying decision rationales of AI models, assisting humans in evaluating the reliability of AI decisions and identifying the optimal strategies to rely on AI recommendations. However, many empirical studies [12–18], which evaluate the effectiveness of current XAI methods for improving people's understanding

38th Conference on Neural Information Processing Systems (NeurIPS 2024).

of the AI model [16, 19, 20] and supporting their calibrated trust in the model [13, 14, 20], have demonstrated that humans often struggle to use the explanations generated by these methods in the optimal way. Thus, despite designers' expectations that XAI methods will positively shape human interaction with AI models, they often fall short of their intended goals, such as fostering appropriate levels of trust and reliance in AI-assisted decision-making. This could be because existing XAI methods do not account for human reactions, making them unadaptive to human cognitive processes. If this is the case, one may naturally wonder if it is feasible to quantitatively model how humans incorporate both AI recommendations and explanations into their decision making process. If so, can the quantitative understanding of human behavior obtained from the learned behavior models be utilized to directly manipulate AI explanations, thereby nudging human decision-makers towards making targeted decisions?

To answer these questions, in this paper, we begin by training behavior models that characterize how AI recommendations and explanations are factored into human decisions based on the collected behavior datasets for various decision making tasks. Utilizing these learned models, we then adjust the AI explanations for different purposes through gradient-based optimizations. Our extensive human experiments across various tasks demonstrated that such human behavior modeling can bring forth both *benefits* and *risks* — when used for benign purposes, the human behavior models can inform the adjustment of AI explanations such that the manipulated explanations could significantly enhance the decision-making performance of human-AI teams in most tasks; however, the same behavior models can also be exploited by adversarial parties for adversarial purposes, such as increasing human decision-makers' biases against a certain protected group and significantly decreasing the fairness level of humans' final decisions across different groups. Finally, examining human perceptions of these manipulated explanations reveals the "*scary*" truth — while human decisions are easily swayed by these altered explanations, individuals generally fail to detect any anomalies in the explanations, underscoring a significant vulnerability in human-AI interaction.

## 2 Related Work

### 2.1 Computational Modeling of AI-assisted Decision Making

There has been a surge of interests among researchers recently in computationally modeling human behavior in AI-assisted decision making [1–5, 21–24]. The goals of these studies for modeling human behavior are diverse, encompassing improving human-AI team performance through intelligent interventions or model recommendation adjustments [6, 25–29], deciding when to present AI powered code suggestion in programming [30], evaluating the utility of AI explanations to improve user understanding of AI model behavior [31–33], and deploying adversarial attacks on AI models to reduce human trust [34]. In this paper, we take a more holistic view and explore whether we can model how humans integrate both AI recommendations and explanations into their decision making process, and what the implications of such behavior modeling are.

### 2.2 Human-centered Evaluation of AI Explanations

With the increasing use of AI technologies as decision aids for humans, a variety of explainable AI techniques have been developed to increase the interpretability of AI models [7–11, 33, 35–37]. To understand the effectiveness of these explanation methods, a growing body of empirical human-centered evaluations have been conducted to examine how AI explanations would affect the ways that humans perceive and interact with AI models [12–18, 31, 32, 38, 39]. These evaluations look into various aspects of impacts of AI explanations, such as the influence on people's trust and reliance on the AI model [13, 14, 20], understanding of AI model [16, 19, 20], and the collaboration performance of the human-AI team [12, 40, 41]. Recently, some research has explored the modification of AI explanations to influence human behavior. For example, Lakkaraju and Bastani [42] demonstrated that handcrafting modifications in AI explanations—such as hiding sensitive features—can mislead human trust in AI models. Another study [43] found that aligning AI explanations with humans' own decision rationales can increase agreement between human decisions and the AI model's predictions. Different from previous work which required intensive handcrafting of AI explanations, in this paper, we explore whether it is possible to directly exploit the computational human behavior models to manipulate AI explanations, even without the access to AI models, with the goal of nudging human decisions towards targeted directions.

# 3  Methodology

## 3.1  Problem Formulation

In this study, we explore the scenario of human-AI collaboration within the context of AI-assisted decision making, and we now formally describe it. Consider a decision making task represented by a $n$-dimension feature vector $\boldsymbol{x} \in \mathcal{R}^n$, and $y$ is the correct decision to make in this task. Specifically, in this study, we focus on decision making tasks with binary choices of decisions, i.e., $y \in \{-1, 1\}$. The AI model's recommendation on the decision task is represented as $y^m = M(\boldsymbol{x})$, $y^m \in \{-1, 1\}$. Following the explainable AI methods like LIME[7] or SHAP[8], the AI model could also provide some "explanations" of its decision, $\boldsymbol{e} = \mathcal{E}(M(\boldsymbol{x}))$, $\boldsymbol{e} \in \mathcal{R}^n$, by showing the contributions of each feature to the decision. With all these information, the human decision maker (DM) needs to make the final decision $y^h \in \{-1, 1\}$ by either accepting or rejecting AI model's decision recommendation $y^m$, which can be characterized by $y^h = \mathcal{H}(\boldsymbol{x}, y^m, \boldsymbol{e})$. The goal of our study is to explore whether we can quantitatively model such decision making process—specifically, $\mathcal{H}(\boldsymbol{x}, y^m, \boldsymbol{e})$—and whether this quantitative understanding of human behavior can be utilized to adjust AI explanations (i.e., change $\boldsymbol{e}$ to $\boldsymbol{e}'$) without accessing to the original AI model $M(\cdot)$, thereby nudging human DMs to make the targeted decision $\hat{y}^h \in \{-1, 1\}$, denoted as $\hat{y}^h = \mathcal{H}(\boldsymbol{x}, y^m, \boldsymbol{e}')$.

## 3.2  Modeling Human Behavior in AI-assisted Decision Making

We first build computational models to characterize how humans integrate both AI recommendations and explanations into their decision process. Following previous works on modeling human behavior in different scenarios of AI-assisted decision making [23, 31], we adopted a two-layer neural network as the structure for modeling the human decision in this study:

$$y^h = \mathcal{H}_{\boldsymbol{w}_h}(\boldsymbol{x}, y^m, \boldsymbol{e}) = \mathcal{H}_{\boldsymbol{w}_h}([\boldsymbol{x}, y^m, \boldsymbol{e}, \boldsymbol{x} \odot \boldsymbol{e}]) \tag{1}$$

The inputs to the behavior model include the task features $\boldsymbol{x}$, the AI model's prediction $y^m$, the AI explanation $\boldsymbol{e}$, and the interaction term between the task features and the AI explanation $\boldsymbol{x} \odot \boldsymbol{e}$ that reflects how humans may redirect their attention to the corresponding features highlighted by the AI explanation. Given the human behavior dataset $\mathcal{D} = \{\boldsymbol{x}_i, y_i^m, \boldsymbol{e}_i, y_i^h\}_{i=1}^N$, we can employ the maximum log-likelihood estimation to learn the behavior model $\mathcal{H}_{\boldsymbol{w}_h}$.

## 3.3  Manipulating AI Explanations through the Behavior Model

We next proceed to explore how the quantitative understanding of human behavior from $\mathcal{H}_{\boldsymbol{w}_h}$ can be utilized to manipulate AI explanations. In particular, given the targeted decision $\hat{y}^h$ for the task instance $\boldsymbol{x}$, we want to identify a new AI explanation $\boldsymbol{e}'$ that maximizes the likelihood that human DMs make the targeted decision $\hat{y}^h$ according to the learned behavior model $\mathcal{H}_{\boldsymbol{w}_h}$. In addition, to prevent the case where the manipulated explanations $\boldsymbol{e}'$ has a very low level of fidelity [10], such as suggesting a recommendation that is inconsistent with the AI model's prediction $y^m$, we also impose a constraint that the new explanation $\boldsymbol{e}'$ should still support the original AI recommendation $y^m$. Since we assume no access to the original AI model, we define $\mathcal{L}_{\text{consistency}}(\boldsymbol{e}, y^m)$ as a measurement of agreement consistency between the manipulated AI explanations and the AI recommendation:

$$\mathcal{L}_{\text{consistency}}(\boldsymbol{e}, y^m) = \begin{cases} 0 & \text{if sign}\left(\sum_i \boldsymbol{e}_i\right) = \text{sign}(y^m), \\ 1 & \text{otherwise.} \end{cases} \tag{2}$$

Together, we use the following optimization problem to manipulate AI explanations:

$$\text{argmin}_{\boldsymbol{e}' \in \mathcal{R}^n} \mathcal{L}_{\text{behavior}}(\mathcal{H}_{\boldsymbol{w}_h}(\boldsymbol{x}, y^m, \boldsymbol{e}'), \hat{y}^h), \text{ subj. to } \mathcal{L}_{\text{consistency}}(\boldsymbol{e}', y^m) \leq 0 \tag{3}$$

where $\mathcal{L}_{\text{behavior}}$ is defined as the cross entropy function. Since exactly solving the above optimization problem is intractable, we used the gradient-based optimization to approximate it:

$$\boldsymbol{e}'_{\theta^{t+1}} = \boldsymbol{e}'_{\theta^t} - \eta \nabla_{\boldsymbol{e}'_\theta}(\mathcal{L}_{\text{behavior}}(\mathcal{H}_{\boldsymbol{w}_h}(\boldsymbol{x}, y^m, \boldsymbol{e}'_\theta), \hat{y}^h) + \lambda \mathcal{L}_{\text{consistency}}(\boldsymbol{e}'_\theta, y^m)) \tag{4}$$

where $\eta$ is the step size, $\lambda$ is the trade-off parameter, and $\boldsymbol{e}'_\theta$ represents the parameterized explanations in the optimization process. We can iteratively optimize manipulated explanations $\boldsymbol{e}'$ until $\mathcal{L}_{\text{behavior}}$ is smaller than a threshold $\tau$ or reach the maximum number of rounds $T$.

# 4 Human Behavior Model Learning

To develop the human behavior model for manipulating AI explanations, we first conduct a human subject experiment to collect human behavior data.

## 4.1 Decision Making Task and AI Assistance

We consider four decision making tasks in this study:

- **Census Prediction (Tabular Data)** [44]: This task was to determine a person's annual income level. In each task, the human DM was presented with a profile with 7 features, including the person's gender, age, education level, martial status, occupation, work type, and working hour per week. The subject was asked to decide whether this person's annual income is higher or lower than $50k for each task. We trained a random forest model to make the income prediction, and the accuracy of the AI model was 76%.

- **Recidivism Prediction (Tabular Data)** [45]: This task was to determine a person's recidivism risk. In each task, the human DM was presented with a profile with 8 features, including their basic demographics (e.g., gender, age, race), criminal history (e.g., the count of prior non-juvenile crimes, juvenile misdemeanor crimes, juvenile felony crimes committed), and information related to their current charge (e.g., charge issue, charge degree). The subject was asked to decide whether this person would reoffend within two years. We trained a random forest model to make the prediction, and the accuracy of the AI model was 62%.

- **Bias Detection (Text Data)** [46]: In this task, the human DM was presented with a text snippet and needed to decide whether it contained any bias. We fine-tuned a BERT [47] model to identify bias in the snippet, and the accuracy of the AI model is 79%.

- **Toxicity Detection (Text Data)** [48]: In this task, the human DM was presented with a text snippet and needed to decide whether it contained any toxic content. We fine-tuned a BERT model to identify the toxic content, and the accuracy of the AI model is 86%.

To understand how people respond to various AI explanations, we employed LIME and SHAP to explain the predictions made by the AI model. Additionally, we augment the LIME or SHAP explanations by either randomly masking out contributions from some features or amplifying contributions of some features (referred to as the "Augmented" explanations) to see how humans react to them. These explanations are provided with AI recommendations together to humans in decision making.

## 4.2 Experimental Procedure

We posted our data collection study on the Prolific [1] to recruit human participants. Upon arrival, we randomly assigned each participant to one of the four decision making tasks and they needed to fill in an initial survey to report their demographic information and their knowledge of AI models and explanations. Participants started the study by completing a tutorial that described the decision making task that they needed to work on. To familiarize participants with the task, we initially asked them to complete five tasks independently without AI assistance. During these training tasks, we immediately provided the correct answer at the end of the task. After the completion of training tasks, participants moved on to the formal tasks. In the formal tasks, participants would receive one type of AI explanations among SHAP, LIME, or Augmented. Specifically, each participant was asked to complete a total of 15 tasks. In each task, participants were provided with the AI prediction and the explanations along with the task instance. They were then required to make their final decisions. Finally, participants were required to complete an exit survey to report their perceptions of the AI explanations they received during the study. They were asked to rate the alignment of AI explanations with their own rationale, as well as the usefulness, transparency, comprehensibility, satisfaction with the provided explanations, and their trust in the AI models, on a 5-point Likert scale. For the detailed survey questions, please refer to Appendix A.3. We offered a base payment of $1.2 and a potential bonus of $1 if the participant's accuracy is above 85%. The study was open to US-based workers only, and each worker can complete the study once.

---

[1]https://www.prolific.com/

Table 1: The number of subjects recruited in data collection for training behavior models, and the average accuracy of the human behavior model in 5-fold cross validation for each task.

| | Census | Recidivism | Bias | Toxicity |
|---|---|---|---|---|
| Number of Participants | 78 | 80 | 72 | 42 |
| Model Accuracy | 0.74 | 0.79 | 0.65 | 0.76 |

### 4.3 Training Results

After collecting data on human behavior, we developed human behavior models for each type of task. For the human behavior models for two textual tasks—**Toxicity Detection** and **Bias Detection**—we employed the pretrained BERT encoder to extract features from the original sentences, which were then used as the task feature $x$ in the human behavior model $H_{w_h}$. We optimized these behavior models using Adam [49] with an initial learning rate of $1e-4$ and a batchsize of each training iteration of 128. The number of training epochs is set as 10. Table 1 shows the number of participants recruited, as well as the average accuracy of the human behavior model evaluated through 5-fold cross validation for each task. We observed that the average accuracy of all human behavior models exceeds 0.65, which is considered to be reasonable. Consequently, we utilized these learned human behavior models to manipulate AI explanations in the following evaluations.

## 5    Evaluation I: Manipulating AI Explanations for Adversarial Purposes

In our first evaluation, we adopted the role of an adversarial party to explore whether they could utilize the learned human behavior model to manipulate AI explanations. The manipulation goal was to nudge human DMs to be biased against certain protected groups in the decision making process. We are particularly interested in comparing the fairness level of human decision outcomes between human DMs who receive original explanations, such as SHAP or LIME, and those who receive manipulated explanations. Notably, all human DMs are provided with the same AI predictions for the same decision making task. Additionally, we also explore differences in human perceptions of original AI explanations versus manipulated AI explanations.

**Evaluation Metrics and Manipulating AI Explanations.** Following previous work [50, 51], we used the false positive rate difference (i.e., *FPRD*) and the false negative rate difference (i.e., *FNRD*) to measure the fairness level of human decision outcomes—the closer these values are to zero, the more fair the decisions are. To manipulate AI explanations and nudge human DMs toward biasing against certain protected groups, we define the targeted human decision $\hat{y}^h$ for each task as follows:

- **Census Prediction**: In this task, we considered a person's sex as the protected attribute. The targeted human decision is defined as $\hat{y}^h = 1$ (indicating a person's annual income exceeds \$50K) when $x_{\text{sex}}$ = male, and $\hat{y}^h = -1$ (indicating a person's annual income does not exceed \$50K) when $x_{\text{sex}}$ = female. The fairness metrics can be computed as $FPRD = FPR_{\text{female}} - FPR_{\text{male}}$, and $FNRD = FNR_{\text{female}} - FNR_{\text{male}}$.

- **Recidivism Prediction**: In this task, we considered the defendant's race as the protected attribute. The targeted human decision is defined as $\hat{y}^h = 1$ (indicating the defendant will reoffend) when $x_{\text{race}}$ = black, and $\hat{y}^h = -1$ (indicating the defendant will not reoffend) when $x_{\text{race}}$ = white. The two fairness metrics can be computed as $FPRD = FPR_{\text{white}} - FPR_{\text{black}}$, and $FNRD = FNR_{\text{white}} - FNR_{\text{black}}$.

- **Bias Detection**: In this task, we divided text snippets into groups based on their political leaning. The targeted human decision is defined as $\hat{y}^h = 1$ (indicating the text is biased) when $x_{\text{leaning}}$ = democratic, and $\hat{y}^h = -1$ (indicating the text is not biased) when $x_{\text{leaning}}$ = republican. The fairness metrics can be computed as $FPRD = FPR_{\text{rep}} - FPR_{\text{dem}}$, and $FNRD = FNR_{\text{rep}} - FNR_{\text{dem}}$.

- **Toxicity Detection**: In this task, we divided text snippets into groups based on the victim of the text. The targeted human decision is defined as $\hat{y}^h = 1$ (indicating the text is toxic) when $x_{\text{victim}}$ = white, and $\hat{y}^h = -1$ (indicating the text is non-toxic) when $x_{\text{victim}}$ = black. The two fairness metrics can be computed as $FPRD = FPR_{\text{black}} - FPR_{\text{white}}$, and $FNRD = FNR_{\text{black}} - FNR_{\text{white}}$.

Table 2: The number of participants we recruited in the evaluation study, categorized according to the type of AI explanation they received and the task they were assigned to.

|  | Census | Recidivism | Bias | Toxicity |
|---|---|---|---|---|
| SHAP | 86 | 89 | 60 | 88 |
| LIME | 65 | 71 | 59 | 85 |
| Adversarially Manipulated | 82 | 92 | 71 | 65 |
| Benignly Manipulated | 77 | 84 | 69 | 46 |

For the Bias Detection and Toxicity Detection tasks, the original datasets [46, 48] provide annotations for the political leanings of the sentences and the targeted victims of the text snippets, respectively. After determining the targeted decision $\hat{y}^h$ for each task instance, we then followed the gradient-based optimization procedure (i.e., Equation 4) to identify the manipulated explanation. We set the step size $\eta$ as $0.01$, the trade-off $\lambda$ as $0.01$, the optimization threshold $\tau$ as $0.1$, and the maximum optimization number $T$ as $100$. For the initial AI explanation $e_{\theta^0}$ at the start of the optimization process, we directly initialize $e'_{\theta^0}$ as $e'_{\theta^0} \sim U(-1, 1)$. We then repeated this optimization process for 5 times and took the average to use in the following human evaluations. For the examples of manipulated explanations, please refer to Appendix B.3.

**Data Collection.** We followed the experimental procedure described in Section 4.2 to collect data on human responses to and perceptions of different AI explanations. We randomly assigned either SHAP or LIME explanations, or the manipulated explanations, to participants. Participants were required to complete 15 tasks with AI model predictions and the assigned explanations (SHAP or LIME or manipulated). We offered a base payment of \$1.2 and a potential bonus of \$1 if the participant's accuracy is above 85%. Table 2 reports detailed statistics of the participants in each task. Below, we analyzed how the manipulated explanations affect fairness level of human decisions and how do humans perceive those explanations.

### 5.1 How do the adversarially manipulated explanations affect fairness level of human decisions?

The fairness levels of participants' decision outcomes under the manipulated explanation, SHAP explanation, and LIME explanation are presented in Figure 1. Visually, it appears that when human DMs are provided with manipulated explanations, both *FPRD* and *FNRD* scores of their decision outcomes tend to deviate more from zero compared to when DMs receive SHAP or LIME explanations.

To examine whether these differences are statistically significant, we conducted regression analyses. Specifically, the focal independent variable was the type of explanation received by participants, while the dependent variables were the participants' *FPRD* and *FNRD* scores. To minimize the impact of potential confounding variables, we included a set of covariates in our regression models, such as participants' demographic background (e.g., age, race, gender, education level), their knowledge of AI explanations, their trust in AI models, and the *FPRD* or *FNRD* scores of the AI model decisions they received in the study. These covariates were selected based on prior HCI research [14, 20, 41] which empirically reveal how characteristics of human DMs may moderate the impacts of AI explanations on human decisions in AI-assisted decision making.

Our regression results indicate that the adversarial party can significantly increase the level of unfairness in human decision outcomes with manipulated explanations through human behavior modeling. Specifically, when examining *FPRD*, we found that participants who received manipulated AI explanations made more unfair decisions compared to those who received SHAP or LIME explanations ($p < 0.05$) in the Census and Recidivism tasks. The difference was marginally significant ($p < 0.1$) in the Toxicity task. When examining *FNRD*, results show that participants who received manipulated explanations made decisions that were significantly more unfair than those who received SHAP or LIME explanations ($p < 0.01$) in the Bias task.

### 5.2 How do humans perceive the adversarially manipulated AI explanations?

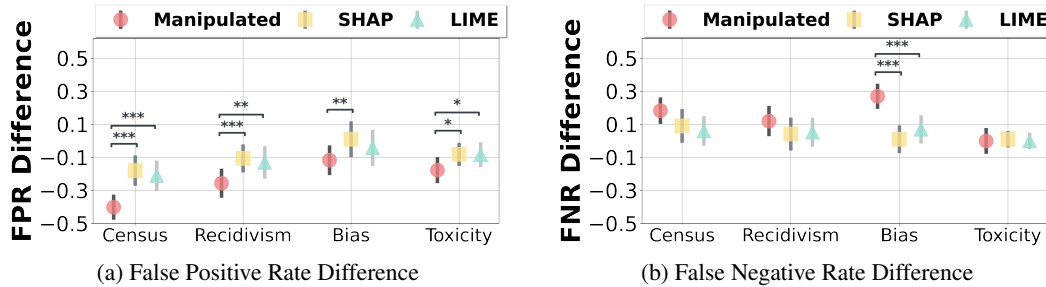

(a) False Positive Rate Difference      (b) False Negative Rate Difference

Figure 1: Comparing *average FPRD* and *FNRD* of the human decision outcomes under the adversarially manipulated explanation, SHAP explanation, or LIME explanation. Error bars represent the 95% confidence intervals of the mean values. *, **, and *** denote significance levels of 0.1, 0.05, and 0.01, respectively. For both *FPRD* and *FNRD*, a value closer to zero indicates that the human decisions are more *fair*.

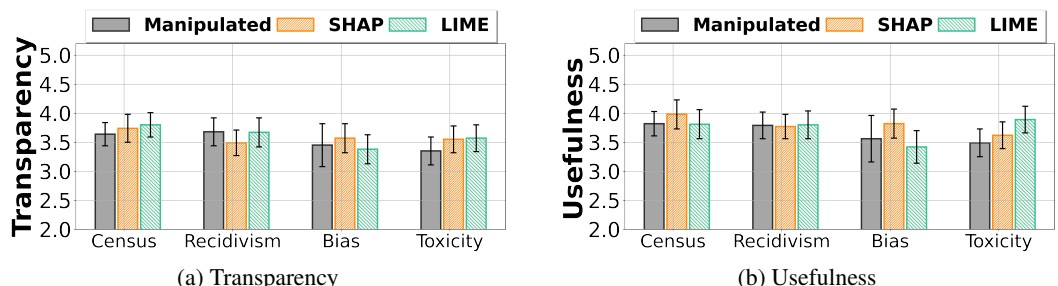

(a) Transparency      (b) Usefulness

Figure 2: Comparing the *average* human perceived transparency and usefulness of the adversarially manipulated explanation, SHAP explanation, and LIME explanation. Error bars represent the 95% confidence intervals of the mean values.

In Section 5.1, we found that the adversarial party can manipulate AI explanations to nudge human DMs toward making more unfair decisions compared to those who received the original AI explanations, aligning with the adversarial party's intentions. To determine whether DMs could detect any abnormalities in the manipulated explanations, we examined how their perceptions of AI explanations varied among manipulated, SHAP, and LIME explanations.

Figures 2a and 2b compare the average perceived transparency and usefulness of three types of AI explanations. Visually, there are no significant differences in how the explanations are perceived by people who received different explanations. We also applied regression models to predict human perceptions of these explanations by accounting for their demographic background (e.g., age, race, gender, education level), their knowledge of AI explanations and their trust in AI models. The regression results indicate that there are no significant differences in the perceived transparency and usefulness of manipulated explanations compared to SHAP or LIME explanations. Similar patterns were observed for perceptions of alignment, comprehensibility, satisfaction, and trust between the manipulated and unmanipulated explanations. While adversarially manipulated explanations significantly influence human decision making behavior, individuals generally do not detect abnormalities in the manipulated AI explanations across most tasks. For further details, please refer to Appendix B.2.

## 6   Evaluation II: Manipulating AI Explanations for Benign Purposes

In the previous section, we found that the adversarial party could use the behavior model to manipulate AI explanations, thereby misleading humans into making unfavorable decisions against specific groups. Naturally, one might wonder could a third party also use behavior models to manipulate AI explanations for benign purposes, such as promoting more appropriate human reliance on AI models? For instance, can manipulated AI explanations lead humans to reject AI recommendations when the

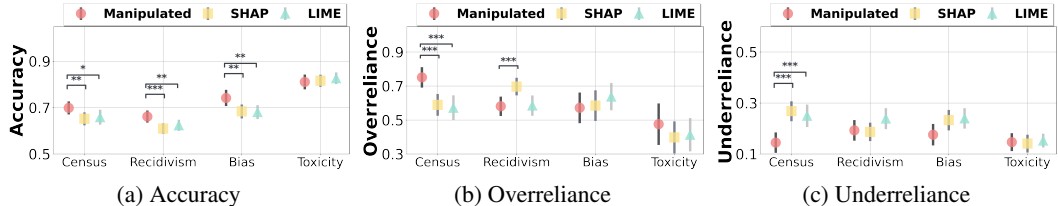

(a) Accuracy  (b) Overreliance  (c) Underreliance

Figure 3: Comparing the *average* accuracy, overreliance, and the underreliance of human decision outcomes under the benignly manipulated explanation, SHAP explanation, or LIME explanation. Error bars represent the 95% confidence intervals of the mean values. *, **, and *** denote significance levels of 0.1, 0.05, and 0.01 respectively.

AI model decision is likely incorrect, and encourage acceptance when the decision is likely correct? We aim to explore the answers to this question in this section.

**Evaluation Metrics and Manipulating AI Explanations**    Following previous work [14, 41, 43], we used the accuracy, underreliance, and overreliance to measure human DMs' appropriate reliance level on AI models. Underreliance refers to the fraction of tasks where the participant's decision was *different* from the AI model's decision when the AI model's decision was correct. Overreliance refers to the fraction of tasks where the participant's decision was the *same* as the AI model's decision when the AI model's decision was incorrect. To manipulate AI explanations for the promotion of more appropriate reliance on AI models, it is necessary to determine the reliability of AI model prediction on each task instance. Recent work [52, 53] has proposed methods to leverage the complementary strengths of humans and AI by combining human independent decisions and AI model, which is often shown to result in more accurate decisions than those made by either humans or AI models alone. Specifically, given the human independent decision $y_{\text{independent}}^h$, the AI model recommendation $y^m$, and the task instance $\boldsymbol{x}$, these methods learn models to combine $y_{\text{independent}}^h$ and $y^m$ to produce a combined result:

$$y_{\text{combine}} = \text{CombineModel}(y_{\text{independent}}^h, y^m, \boldsymbol{x}) \tag{5}$$

To see whether $y_{\text{combine}}$ can yield better decisions compared to AI alone or human alone, we evaluated various combination models including the human-AI combination method [53] and several truth inference methods [54–57] used in crowdsourcing. Our results showed that the human-AI combination method [53] generally outperformed AI solo and independent human decision, as well as other combination methods. Thus, $y_{\text{combine}}$ produced by the human-AI combination method [53] is defined as the targeted decision $\hat{y}^h$ for manipulating AI explanations.  For detailed information on the evaluations of each combination method, please refer to the Appendix C.1. We again followed Equation 4 to manipulate the AI explanations. We set the step size $\eta$ as 0.01, the trade-off $\lambda$ as 0.01, the optimization threshold $\tau$ as 0.1, and the maximum optimization number $T$ as 100, and the initial AI explanation $\boldsymbol{e}_{\theta^0}'$ at the start of the optimization process is initialized as $\boldsymbol{e}_{\theta^0}' \sim U(-1, 1)$. We repeated this optimization process for 5 times and took the average to use in the following experiments. For the examples of manipulated explanations, please refer to Appendix C.4.

**Data Collection.**    We recruited participants from Prolific once again to collect behavioral data under the benignly manipulated explanations, following the experimental procedure described in Section 4.2. We offered a base payment of \$1.2 and a potential bonus of \$1 if the participant's accuracy is above 85%. Table 2 shows the detailed statistics of the participants we recruited for each task. Subsequently, we analyzed whether the benignly manipulated explanations can promote appropriate reliance of human DMs on AI models, as well as their perceptions of these AI explanations.

### 6.1 Can benignly manipulated explanations promote appropriate reliance of human DMs on AI models?

Figures 3a, 3b, and 3c compare the average accuracy, overreliance, and underreliance of human decision outcomes under manipulated, SHAP, and LIME explanations, respectively. It is clear that providing human DMs with manipulated AI explanations leads to an increase in the accuracy of their decision outcomes for most of tasks. We subsequently conducted regression analyses to determine

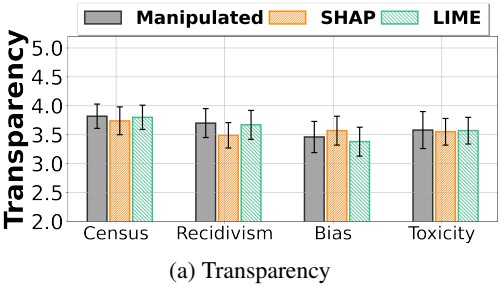
(a) Transparency

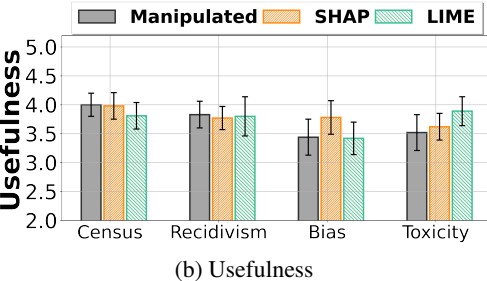
(b) Usefulness

Figure 4: Comparing the *average* human perceived transparency and usefulness of the benignly manipulated explanations, SHAP explanations, and LIME explanations. Error bars represent the 95% confidence intervals of the mean values.

whether these differences are statistically significant. The regression models incorporated a set of covariates, including participants' demographic backgrounds (e.g., age, race, gender, education level), their knowledge of AI explanations, their trust in AI models, and the accuracy of the AI models. The regression results indicate that in the Census, Recidivism, and Bias tasks, substituting SHAP or LIME explanations with manipulated explanations significantly improves the accuracy of human-AI team. In contrast, for the Toxicity task, we observed no statistical difference, which could potentially be attributed to the high competence of humans in solving this task (e.g., when presented with SHAP or LIME explanations, the average decision accuracy of participants already exceeds 0.8, leaving limited room for further improvement).

### 6.2 How do humans perceive benignly manipulated AI explanations?

In Section 5.2, we observed that it is challenging for humans to detect abnormalities in the adversarially manipulated explanations, even though they are unconsciously influenced by the manipulated explanations to make more unfair decisions. In this section, we revisit this question to investigate into whether humans' perceptions of the manipulated explanations change, when they are manipulated for benign purposes. Figures 4a and 4b compare the average human perceived transparency and usefulness of the benignly manipulated explanations, SHAP explanations, and LIME explanations. Regression analyses reveal no statistically significant differences among the perceived transparency and usefulness of these three types of explanations. Similar trends were observed for other perceptual aspects of explanations, including perceived alignment, comprehensibility, satisfaction, and trust. For further results, please refer to Appendix C.3.

## 7 Conclusion and Limitations

In this paper, we explore whether we can quantitatively model how humans incorporate both AI recommendations and explanations into their decision making process, and whether we can utilize the quantitative understanding of human behavior obtained from these learned models to manipulate AI explanations for both adversarial and benign purposes. Our extensive experiments across various tasks demonstrate that human behavior can be easily influenced by these manipulated explanations toward targeted outcomes, regardless of the intent being benign or adversarial. Despite the significant influence of these falsified explanations on human decisions, individuals typically fail to detect or recognize any abnormalities. Our study has several limitations. For example, it focuses on modeling and manipulating score-based explanations. Further research is needed to explore how to model how humans incorporate other types of explanations, such as example-based and rule-based explanations, and how these can be manipulated to influence human behavior as observed with score-based explanations in our study. Additionally, our study was limited to decision making tasks involving tabular and textual data, which are naturally suited to score-based explanations. Further explorations are needed to extend these findings to decision tasks with other data types (e.g., images).

## Ethical Consideration

This study was approved by the Institutional Review Board of the authors' institution. Through our findings, we aim to draw the community's attention to the ease with which third parties can manipulate AI explanations with the learned behavior models to influence human decision making. Users often lack the ability to accurately and appropriately interpret the AI explanations presented to them, yet their decision behavior is easily swayed by the manipulated AI explanations. Our findings highlight the critical importance of securing human-AI interaction data to prevent the misuse of human behavior models derived from it. Additionally, there is an urgent need to ensure that AI explanations provided to humans are more secure and inherently benign. Moreover, providing pre-education is essential to assist humans in establishing a proper understanding of AI explanations, which may potentially mitigate the risks of manipulation.

In addition, our experiments are based on datasets that are publicly available; "correct" decisions for the tasks in these datasets are generally considered as recording the real-world ground truth. While these datasets are not intentionally biased toward any specific groups, we acknowledge that there might be implicit biases introduced to these datasets during the curation process, which are beyond our control. Importantly, we note that we made no alterations to the datasets that would introduce additional bias in our experiment.

## Acknowledgments

We thank the support of the National Science Foundation under grant IIS-2229876 and IIS-2340209 on this work. Any opinions, findings, conclusions, or recommendations expressed here are those of the authors alone.

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

Table A.1: The average hourly payment received by participants in our study across four tasks. In the row "Number of Participants", the number in parentheses indicates the number of invalid participants who did not pass the attention check questions.

| | Recidivism | Census | Bias | Toxicity |
|---|---|---|---|---|
| Number of Participants | 336 (16) | 310 (25) | 259 (20) | 286 (17) |
| Average Working Time (Minute) | 6.67 | 6.25 | 6.98 | 6.34 |
| Hourly Payment (Base) | $10.8 | $11.7 | $10.2 | $11.3 |
| Hourly Payment (Base + Bonus) | $11.9 | $11.8 | $11.4 | $14.4 |

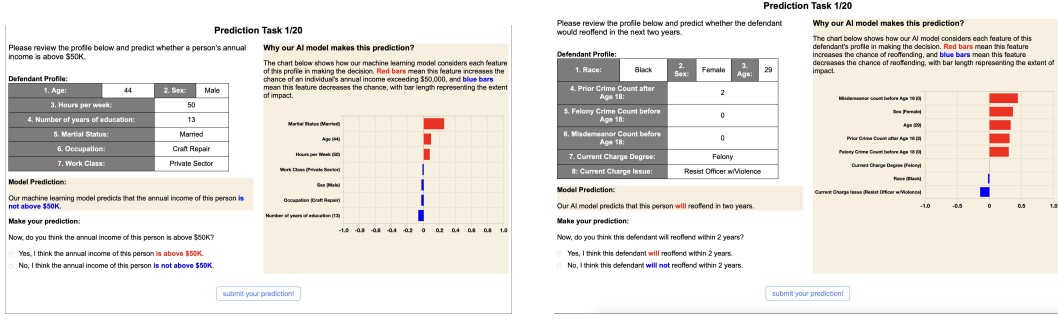

(a) Census prediction          (b) Recidivism prediction

Figure A.1: The task interfaces for the census prediction and recidivism prediction.

# A  The Design of Human Study (Additional Details)

## A.1  Compensation Details

To determine the appropriate payment level for each type of task, we first conducted a preliminary study to estimate the time workers might spend on the tasks. Our pilot study indicated that a base payment of $1.2 per task translates to an approximate hourly rate of $10. To provide greater transparency about the compensation received by participants in our formal study, Table A.1 summarizes the average hourly payment and the average time spent on each task.

## A.2  Task Interfaces

Figure A.1a, A.1b, A.2a, and A.2b show the interfaces participants used in the **Census Prediction**, **Recidivism Prediction**, **Bias Detection**, and **Toxicity Detection** tasks, respectively.

## A.3  Exit Survey Questions

In the study, after the main tasks, the participants need to fill in an exit survey to report their perceptions of presented AI explanations. The survey questions are detailed as:

- **Alignment**: On a scale of 1-5, how well do you think the explanations align with your understanding of the problem?
- **Usefulness**: On a scale of 1-5, how useful are the explanations in helping you make decisions?
- **Transparency**: On a scale of 1-5, how well do you think the explanations reveal the AI model's decision making process?
- **Comprehensibility**: On a scale of 1-5, how easy is it for you to understand the explanations?
- **Satisfaction**: On a scale of 1-5, how satisfied are you with the explanations provided by the AI model?
- **Trust**: On a scale of 1-5, would you trust the AI model's prediction or decision based on the explanations?

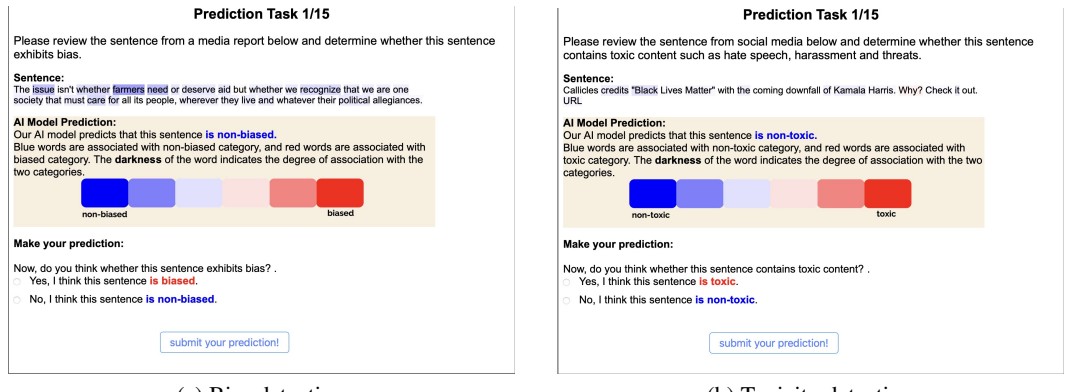

(a) Bias detection          (b) Toxicity detection

Figure A.2: The task interfaces for the bias detection and the toxicity detection.

Table B.1: Agreement between the sum of feature importance in explanations and AI predictions, measured in terms of the Pearson correlation coefficient.

|  | Census | Recidivism | Bias | Toxicity |
|---|---|---|---|---|
| SHAP | 0.88 | 0.97 | 0.88 | 0.91 |
| LIME | 0.41 | 0.40 | 0.76 | 0.78 |
| Adversarially Manipulated | 0.57 | 0.38 | 0.73 | 0.68 |
| Benignly Manipulated | 0.87 | 0.68 | 0.89 | 0.89 |

# B   Evaluation I: Manipulating AI Explanations for Adversarial Purposes (Additional Results)

## B.1   Visual Consistency of Explanations

Table B.1 compares the agreement between the sum of feature importance in explanations and AI predictions, the Pearson correlation coefficients, measured in terms of Pearson correlation coefficient, for adversarially manipulated, LIME, and SHAP explanations, respectively. We observed that the visual consistency of the manipulated explanations is lower than that of SHAP but very close to that of LIME.

## B.2   Human Perceptions of Explanations

Figures B.1a to B.1d compare the average human perceived alignment, comprehensibility, satisfaction with the provided explanations ,and the trust in the AI models under the under the adversarially manipulated explanation, SHAP explanation, or LIME explanation. In general, our findings indicate that there are no significant differences in people's perceptions of the three explanations across four tasks, with the exceptions for the alignment and trust in the **Toxicity** task. Specifically, for the **Toxicity** task, participants perceived LIME explanations as aligning more closely with their own rationales than the adversarially manipulated explanations, with a marginally significant difference ($p < 0.1$). Furthermore, participants reported significantly greater trust in the AI models accompanied by LIME explanations compared to those with adversarially manipulated explanations ($p < 0.01$).

## B.3   Examples of Manipulated Explanations

Figures B.2 to B.5 show the visual comparisons of adversarially manipulated, LIME, and SHAP explanations for the Census Prediction task, Recidivism Prediction task, Bias Detection task, and Toxicity Detection task, respectively.

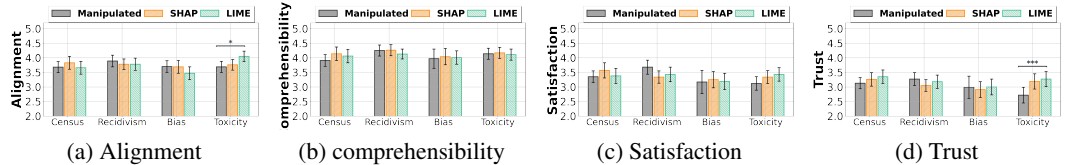

(a) Alignment  (b) comprehensibility  (c) Satisfaction  (d) Trust

Figure B.1: Comparing the *average* human perceived alignment, comprehensibility, satisfaction with the provided explanations, and the trust in the AI models under the adversarially manipulated explanation, SHAP explanation, or LIME explanation. Error bars represent the 95% confidence intervals of the mean values. * and *** denote significance levels of 0.1 and 0.01 respectively.

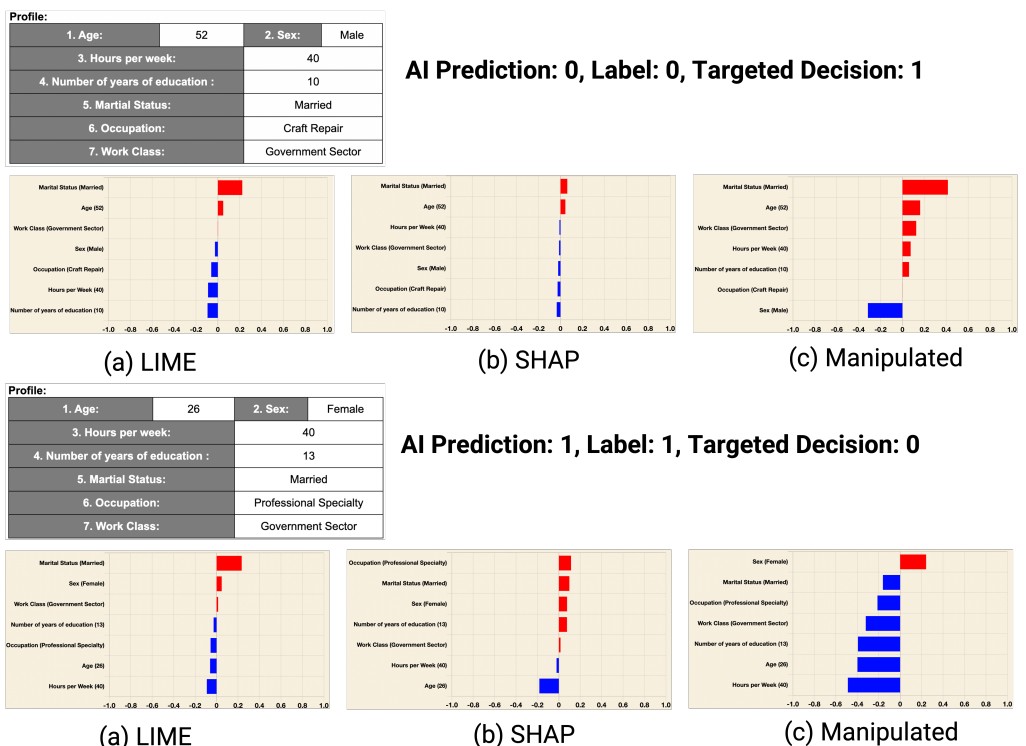

Figure B.2: The visual comparisons of adversarially manipulated, LIME, and SHAP explanations for the Census Prediction task.

# C   Evaluation II: Manipulating AI Explanations for Benign Purposes (Additional Results)

## C.1   Combining Human Decisions and AI Predictions

In the main paper, we aim to benignly manipulate AI explanations to encourage human DMs to rely more appropriately on AI models. Following previous research [52, 53], we combined independent human decisions with AI model predictions to determine the targeted decision for each task instance. We evaluated the human-AI combination method [53] and several truth inference methods used in crowdsourcing for truth discovery. We detailed the process of evaluation below.

Table C.1: The average accuracy of the independent human behavior model through 5-fold validation for each task.

|  | Census | Recidivism | Bias | Toxicity |
|---|---|---|---|---|
| Accuracy | 0.81 | 0.84 | 0.62 | 0.79 |

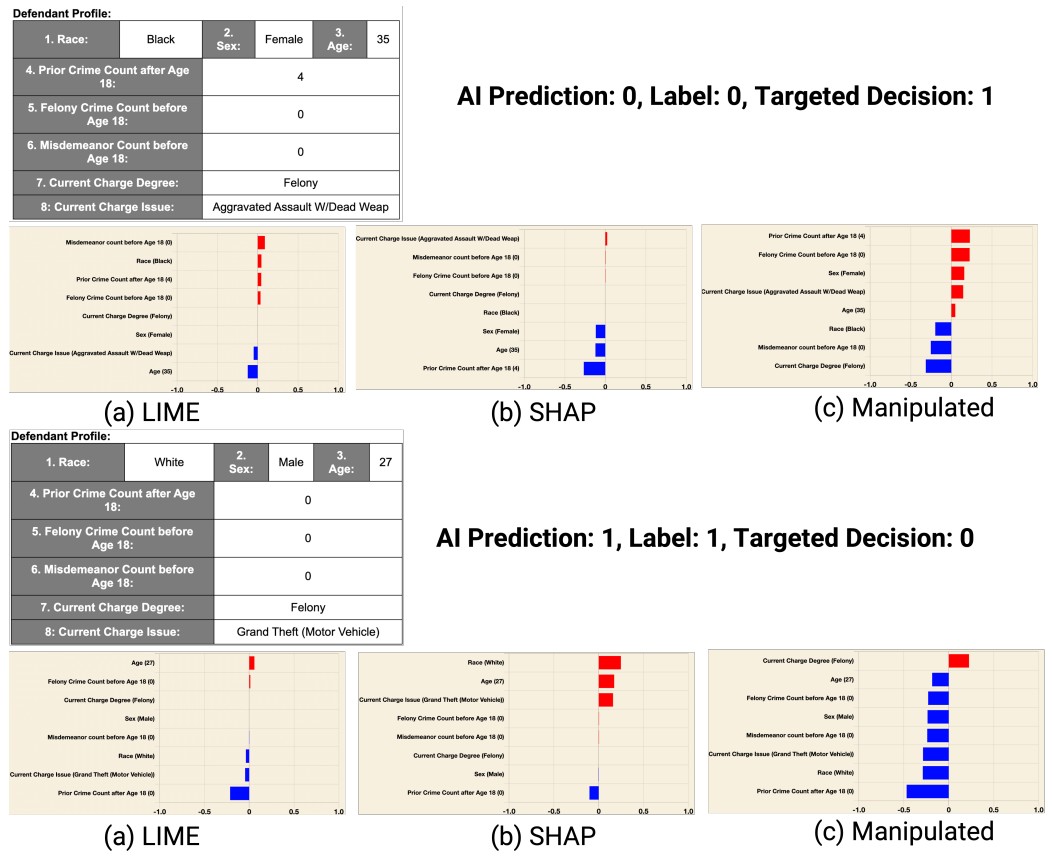

Figure B.3: The visual comparisons of adversarially manipulated, LIME, and SHAP explanations for the Recidivism Prediction task.

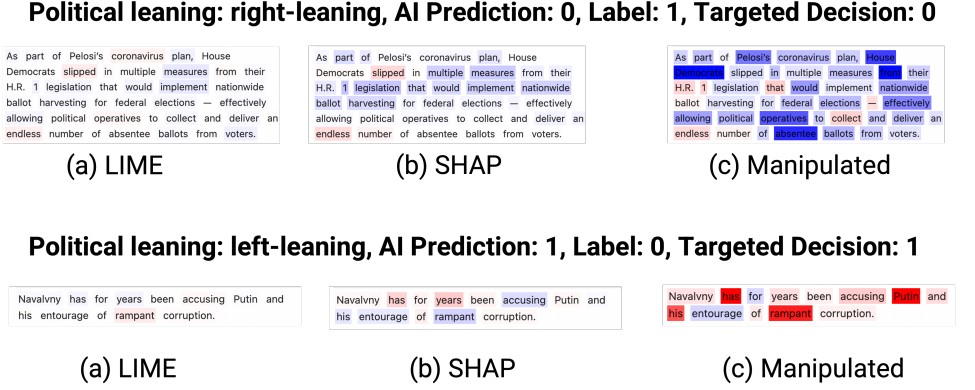

Figure B.4: The visual comparisons of adversarially manipulated, LIME, and SHAP explanations for the Bias Detection task.

**Simulating Human Independent Decision.** To understand how humans independently make decisions on each task instance, we first conducted a study again on the Prolific to collect independent human decision behavior data across four tasks. We recruited 40 participants for each task. Each recruited participant needed to complete 15 tasks. With the collection of human behavior data, we then fitted two-layer neural networks to simulate human independent decision behavior. For **Toxicity Detection** and **Bias Detection** textual tasks, we used the pretrained BERT encoder to initially extract features from the original sentences as the input to the independent behavior models. We optimized these independent behavior models using Adam [49] with an initial learning rate of $1e - 4$ and

**Victim: Black, AI Prediction: 0, Label: 0, Targeted Decision: 0**

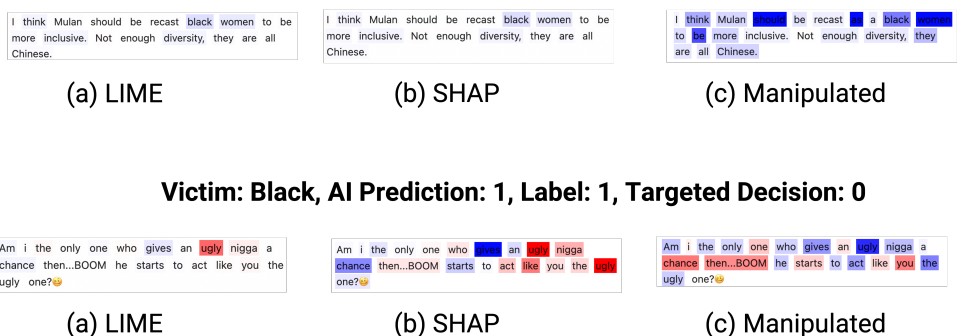

(a) LIME        (b) SHAP        (c) Manipulated

**Victim: Black, AI Prediction: 1, Label: 1, Targeted Decision: 0**

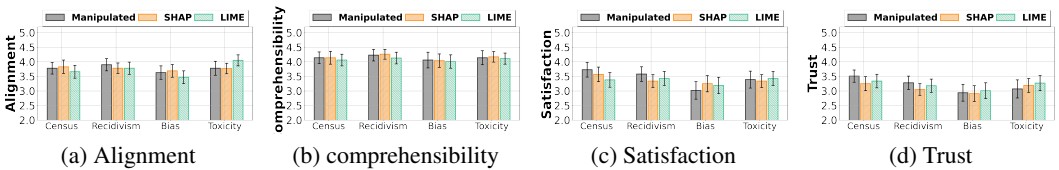

(a) LIME        (b) SHAP        (c) Manipulated

Figure B.5: The visual comparisons of adversarially manipulated, LIME, and SHAP explanations for the Toxicity Detection task.

(a) Alignment    (b) comprehensibility    (c) Satisfaction    (d) Trust

Figure C.1: Comparing the *average* human perceived alignment, comprehensibility, satisfaction with the provided explanations, and the trust in the AI models under the benignly manipulated explanation, SHAP explanation, or LIME explanation. Error bars represent the 95% confidence intervals of the mean values.

a batchsize of each training iteration of 128. The number of training epochs is set as 10. The average accuracy of 5-fold validation for each model is reported in Table C.1, which we found to be satisfactory. We then utilized these fitted models to simulate independent human decisions $y^h_{\text{independent}}$ in the human-AI combination process to determine the potentially better decisions.

**Comparing Combination Performance.** We consider the human + AI combination method [53] and a few truth inference methods in crowdsourcing as baselines in the evaluation, including GLAD [55], CATD [56], LFC [57], EM [54], and MV [54]. These methods combine the human independent decisions $y^h_{\text{independent}}$ predicted by the fitted independent human behavior models and AI model recommendations $y^m$ to produce combined decisions $y_{\text{combine}}$. The accuracy of each method on holdout task pools for each task to be used in the subsequent evaluation is reported in Table C.2. In general, we found that human + AI combination method outperforms other baselines. By integrating human decisions with AI predictions, this method shows superior performance to either AI solo or human solo across all four tasks. Consequently, we used the combined decisions $y_{\text{combine}}$ from the human + AI combination method as the targeted decision $\hat{y}^h$ in subsequent experiments to manipulate explanations.

## C.2 Visual Consistency of Explanations

Table B.1 compares the agreement between the sum of feature importance in explanations and AI predictions, the Pearson correlation coefficients, measured in terms of Pearson correlation coefficient, for benignly manipulated, LIME, and SHAP explanations, respectively. We observed that the visual consistency of the manipulated explanations is very close to that of SHAP and higher than that of LIME.

## C.3 Human Perceptions of Explanations

Figures C.1a to C.1d compare the average human perceived alignment, comprehensibility, satisfaction with the provided explanations ,and the trust in the AI models under the under the adversarially

Table C.2: The accuracy of each method on the holdout task pools, used in following experiments to manipulate AI explanations. The best result in each row is highlighted in **bold**.

| | Human Solo | AI Solo | Human + AI [53] | GLAD [55] | CATD [56] | LFC [57] | EM [54] | MV [54] |
|---|---|---|---|---|---|---|---|---|
| **Census** | 0.61 | 0.73 | **0.76** | 0.62 | 0.74 | 0.61 | 0.60 | 0.69 |
| **Recidivism** | 0.54 | 0.58 | 0.61 | 0.59 | 0.61 | 0.58 | 0.59 | **0.62** |
| **Bias** | 0.55 | 0.80 | **0.81** | 0.66 | 0.65 | 0.67 | 0.69 | 0.66 |
| **Toxicity** | 0.76 | **0.86** | **0.86** | 0.78 | 0.79 | 0.81 | 0.77 | 0.82 |

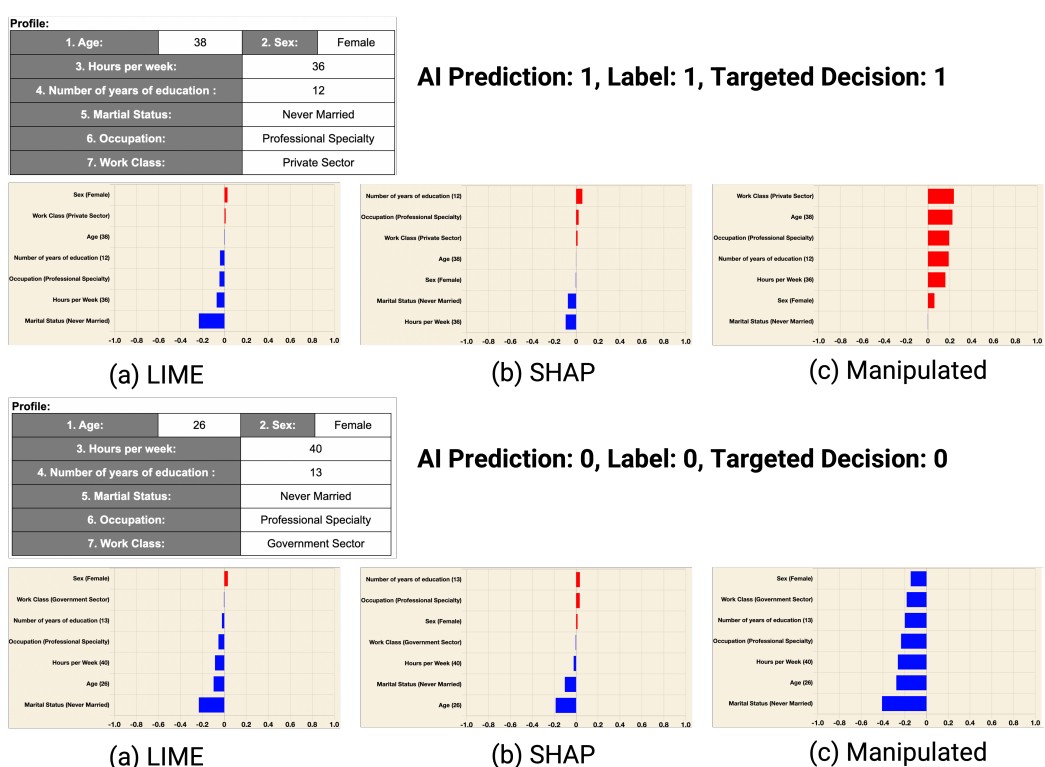

Figure C.2: The visual comparisons of benignly manipulated, LIME, and SHAP explanations for the Census Prediction task.

manipulated explanation, SHAP explanation, or LIME explanation. We found that there are no significant differences in people's perceptions of the three explanations across four tasks on these aspects.

## C.4  Examples of Manipulated Explanations

Figures C.2 to  C.5 show the visual comparisons of benignly manipulated, LIME, and SHAP explanations for the Census Prediction task, Recidivism Prediction task, Bias Detection task, and Toxicity Detection task, respectively.

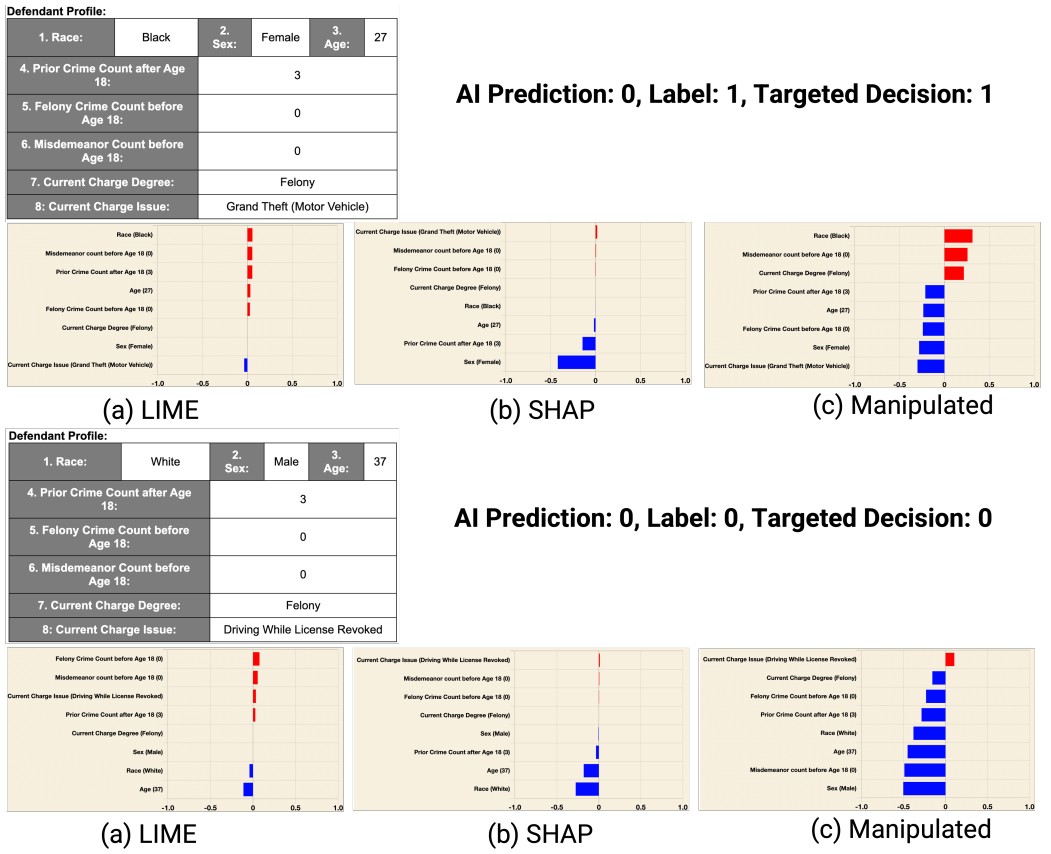

Figure C.3: The visual comparisons of benignly manipulated, LIME, and SHAP explanations for the Recidivism Prediction task.

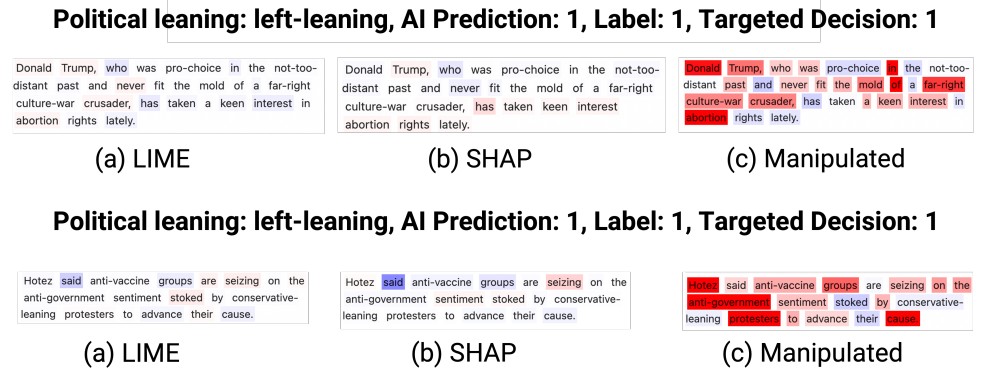

Figure C.4: The visual comparisons of benignly manipulated, LIME, and SHAP explanations for the Bias Detection task.

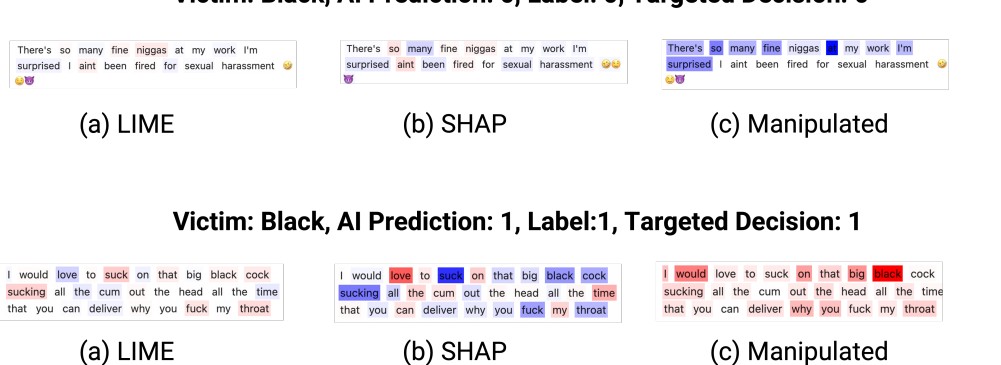

Figure C.5: The visual comparisons of benignly manipulated, LIME, and SHAP explanations for the Toxicity Detection task.

