# OpenReview forum: "Utilizing Human Behavior Modeling to Manipulate Explanations in AI-Assisted Decision Making: The Good, the Bad, and the Scary"
_NeurIPS.cc/2024/Conference — NeurIPS 2024 poster_

### Official Review · Reviewer_9s1B · 2024-06-18

**Soundness:** 3
**Presentation:** 3
**Contribution:** 3
**Rating:** 7
**Confidence:** 2

**Summary:**

This paper addresses the manipulation of explanations in AI-assisted decision-making, presenting a comprehensive study that explores how human behavior models can be used to adjust explanations provided by AI systems. The aim is to understand if these manipulations can nudge decision-makers towards specific outcomes, which might be either beneficial or malicious.

**Strengths:**

1. The paper's focus on quantitatively modeling human behavior to manipulate AI explanations is highly novel and impactful. This approach not only extends the current understanding of AI-human interactions but also opens new avenues for both enhancing and securing AI-assisted decision-making systems.

2.  The experiments conducted across various decision-making tasks provide a robust validation of the proposed models. The inclusion of both adversarial and benign manipulations allows for a balanced view of the potential impacts of this technology.

**Weaknesses:**

1. The scope of the experiments is restricted to tasks such as census and recidivism prediction, which may not adequately represent the complexities and stakes of decision-making environments in sectors like healthcare or finance. Expanding the range of tasks to include high-stakes decision-making could improve the generalizability of the results.
2. The behavior model does not account for the inherent variability and noise in human decision-making, potentially oversimplifying the complexities of real-world human-AI interactions.
2. The data and code are not immediately available for replication and further study, which could hinder the verification of the results and the advancement of the research.

**Questions:**

I still have a positive view of this work, and have the following questions:
1. Could the authors please provide more detail on which covariates were found to be significant in the regression analyses? It would be helpful to understand not only which variables significantly impacted the model outcomes but also how they were selected and their relative influence on the dependent variables.
2. Regarding the use of $\sum_i e_i$ in Eq. 2 for evaluating consistency, I wonder why this summation was chosen over other potential methods. Could the authors clarify the rationale behind this choice?

**Limitations:**

1. Score-based explanations may not be adequate. Incorporating Large Language Models could introduce more versatile and comprehensive explanations.
2. Include more diverse datasets, possibly extending to more complex data types like images or videos, to test the robustness of the behavior models across different AI applications.

---

> ### Author Rebuttal · Authors · 2024-08-05
>
> Thanks for the review! Below we address your questions.
>
> >  Could the authors please provide more detail on which covariates were found to be significant in the regression analyses? It would be helpful to understand not only which variables significantly impacted the model outcomes but also how they were selected and their relative influence on the dependent variables.
>
> - The selection criteria for the covariates, including participants' demographic information, their knowledge of AI explanations, and their trust in AI models, were based on prior HCI research [1,2,3], which empirically examines how AI explanations impact human decisions in AI-assisted decision-making and how these factors might influence that impact.
>
> - In our regression analysis, we observed an interesting trend across all four types of decision tasks, particularly when AI explanations were manipulated for adversarial purposes. Participants with high trust in AI systems consistently rated the manipulated explanations higher in terms of alignment, satisfaction, usefulness, etc. Similarly, participants with greater knowledge of AI explanations exhibited the same trend as those with high trust, suggesting that individuals more familiar with AI systems might be more vulnerable to these manipulations.
>
> We will include the full linear regression results as tables in the manuscript and add a discussion on these findings to provide further insight.
>
> > Regarding the use of  in Eq. 2 for evaluating consistency, I wonder why this summation was chosen over other potential methods. Could the authors clarify the rationale behind this choice?
>
> The choice of the summation form for consistency constraints in Equation 2 is primarily motivated by the alignment with SHAP explanations, which emphasize local accuracy. This property ensures that the sign of the sum of all feature contributions from SHAP can match the sign of the model’s output. Additionally, in our setting, we assume that third parties do not have access to the AI model when manipulating AI explanations. Therefore, alternative metrics that would require probing the model’s outputs with perturbed inputs to evaluate the consistency of explanations are challenging to implement in this study.
>
> > No code available
>
> We will release the collected human behavior data, along with the code for training the human behavior model and manipulating AI explanations, upon the acceptance of our paper.
>
>
> [1] Lai, Vivian, Han Liu, and Chenhao Tan. "" Why is' Chicago'deceptive?" Towards Building Model-Driven Tutorials for Humans." Proceedings of the 2020 CHI Conference on Human Factors in Computing Systems. 2020.
>
> [2] Zhang, Yunfeng, Q. Vera Liao, and Rachel KE Bellamy. "Effect of confidence and explanation on accuracy and trust calibration in AI-assisted decision making." Proceedings of the 2020 conference on fairness, accountability, and transparency. 2020.
>
> [3] Wang, Xinru, and Ming Yin. "Are explanations helpful? a comparative study of the effects of explanations in ai-assisted decision-making." Proceedings of the 26th International Conference on Intelligent User Interfaces. 2021.

---

> > ### Comment · Reviewer_9s1B · 2024-08-09
> > **Thank you for the responses**
> >
> > Thank you to the authors for their responses. Most of my questions have been addressed. After considering your responses and the feedback from other reviewers, I will maintain my evaluation.

---

### Official Review · Reviewer_1Xcr · 2024-07-08

**Soundness:** 3
**Presentation:** 3
**Contribution:** 2
**Rating:** 3
**Confidence:** 3

**Summary:**

This paper proposes to train a computational model to predict how humans would respond to model predictions and their explanations to make the final decision. Using this model, the authors then demonstrate that it could be used to manipulate explanations for both good and bad purposes, specifically to steer human predictions toward the label that is likely to be correct, or make human decisions intentionally biased. Furthermore, the humans have very little idea that the explanations have been manipulated in both cases.

**Strengths:**

This paper focuses on an important topic: the role of explanations in human decision making.

The presentation is generally clear and the writing structure is good.

Extensive experiments are conducted to demonstrate the main arguments of the paper.

**Weaknesses:**

1. I noticed that the user study payment is only \\$1.2 base pay with a potential bonus of \\$1.0 (Sec. 4.2). The study consists of a tutorial, 5 predictions without AI assistance, 15 predictions with AI prediction and explanations, and an exit survey. Given that this study is deployed to US-based participants, the compensation is extremely meager: even at the federal minimum wage of \\$7.25 per hour, the \\$1.2 base pay would be equivalent to 10 min of work, which, given the user interface of Fig. A.1 and A.2, is extremely unlikely to be enough for all the tasks. Thus, I have serious concern about the ethics of the study, despite its IRB approval, and thus decide to request additional ethics reviewers for this paper.

2. The use of computational model for human behavior is not novel, and the applications to human decision manipulation seems quite straightforward.

3. Furthermore, it would be helpful have some additional analysis on the learned computation model itself, maybe with the help of various interpretability tools. For example, when is the human prediction most likely influenced by the provided explanation, and in what way? These quantitative and qualitative insights could be helpful to understand human behaviors better.

4. For the "benign" use case of improving human model prediction, the authors looked at cases "when the AI model decision is likely incorrect" (Line 285). How is this "likely incorrect" determined? And if we know when the AI prediction is likely incorrect, why can't we simply fix/patch the AI model directly? The authors demonstrated that the human performance is better after explanation manipulation, compared to the original explanation in Fig. 3, but is the human prediction better than the "fixed AI model" performance?

**Questions:**

See weaknesses. In addition, the reliance metrics of Fig. 3 are not defined here, and the readers need to search the referenced papers for definitions.

**Limitations:**

Yes.

---

> ### Author Rebuttal · Authors · 2024-08-05
>
> Thank you for your review! We noticed that you are particularly concerned with the ethics of this study. We hope the clarifications below satisfactorily address these issues, and we are open to discussing any further concerns you may have. We are fully committed to ensuring that our work meets all ethical requirements.
>
> >  I noticed that the user study payment is only \\$1.2 base pay with a potential bonus of \\$1.0 (Sec. 4.2). The study …..
>
> Table 1: The average hourly payment received by participants in our study across four tasks. In the row "Number of Workers," the number in parentheses indicates the number of invalid participants who did not pass the attention check questions.
>
> |                               | Recidivism |   Census  |    Bias   | Toxicity |
> |:-----------------------------:|:----------:|:---------:|:---------:|:--------:|
> |       Number of Participants       |  336 (16)  | 310 (25)  | 259 (20)  | 286 (17) |
> |          Average Working Time (minute)         |    6.67    |    6.25  |    6.98   |   6.34   |
> |     Hourly Payment (Base)     |    $10.8   |   $11.7   |   $10.2   |  $11.36  |
> | Hourly Payment (Base + Bonus) |    $11.9   |   $11.8   |   $11.4   |   $14.4  |
>
> - To determine the appropriate payment level for each task, we first conducted a preliminary study to estimate the time workers might spend on the tasks. Our pilot study indicated that a base payment of \\$1.2 per task translates to an approximate hourly rate of \\$10. To provide greater transparency about the compensation received by participants in our formal study, Table 1 summarizes the average hourly payment and the average time spent on each task. As shown in Table 1, the average base hourly payment across the four tasks exceeds \\$10 per hour, and the average hourly payment including bonus is close to \\$12 per hour, both of which are well above the federal minimum wage. In addition, to minimize any potential negative effects of the manipulation, we provided a debrief session through the Prolific system after participants completed the experiment. This session clarified that the provided explanations were intentionally manipulated to influence their decisions.
>
> - In addition, to ensure the quality of participant responses, we included two attention check questions in our study, where participants were required to select a pre-specified answer. These attention checks were randomly inserted among 15 formal tasks. Only participants who passed both attention checks were considered valid for our analysis.
> Lastly, we acknowledge that conducting an online study presents challenges in ensuring that participants maintain full attention throughout the tasks. We will include a discussion of this limitation in our manuscript.
>
> > The use of computational model for human behavior is not novel, and the applications to human decision manipulation seems quite straightforward.
>
>
> While much research has focused on modeling human behavior in AI-assisted decision making, our study is among the first to explore and answer whether we can model the impact of AI predictions with explanations on human decisions. Furthermore,  informed by the learned human behavior models, we further investigate the possibility to  manipulate AI explanations, aiming to advance our understanding of what roles AI explanations play in human decisions. Through our initial exploration of manipulating AI explanations to influence human decision making, we seek to lay a base for demonstrating the potential to strategically manipulate information presented to humans with behavior modeling, thereby impacting their decisions.
>
> > Furthermore, it would be helpful have some additional analysis on the learned computation model itself, maybe with the ….
>
> Thanks for your suggestion! We used SHAP to provide explanations on how task features, AI model predictions, and explanations impact human decisions based on our behavior models. Through our SHAP analysis, we first observed that AI model predictions influence human decisions, often leading them to align with the AI's predictions. In addition, we found that, especially when explanations are manipulated for adversarial purposes, model-based manipulation often intentionally retains certain features as protective to mislead humans into making biased decisions. For example, when predicting whether a person's income is below \\$50k, if the person individual is a female, the manipulated explanation might present gender as a positive contributor to earning above \\$50k, and vice versa for males. If we intentionally change the manipulated contribution of gender to be negative for females, we observed that the predicted probability of this female's income being below $50k by our decision model decreases. We also observed similar trends in recidivism prediction when targeting the 'Black’ race. This phenomenon suggests that when humans make decisions, they may focus intentionally on sensitive features. When AI explanations indicate traces of unfairness in these features, people may naturally make decisions to counteract this perceived bias. And we will include the discussion in the manuscript.
>
> > Continued on the next comment
> .

---

> ### Author Response · Authors · 2024-08-05
> **Rebuttal (continued)**
>
> > For the "benign" use case of improving human model prediction, the authors looked at cases ...
>
>
> - To determine when the AI model's decision is likely correct, we followed an established approach [1], which combines AI predictions with independent human decisions to leverage the complementary strengths of both AI and human judgment. Our evaluation revealed that the combined decision's accuracy on the test set was higher than the accuracy of either the AI model or human judgment alone. Therefore, we used this combined prediction to evaluate when the AI model's decision is likely correct. For more details on this evaluation, please refer to Appendix C.1.
>
> - In AI-assisted decision making, particularly in critical contexts, it is crucial that humans remain responsible for the final decision and its consequences. In addition, simply making AI predictions more accurate may not be enough, because humans could still rely on a highly accurate AI very inappropriately, resulting in low team performance.  Our study, therefore,  is not to enhance AI predictions but to improve the performance of human decision makers as they interact with AI models. In fact, during our evaluation, we found that presenting manipulated explanations can significantly reduce human underreliance—where humans fail to rely on the AI model when it is correct—and thus improve human-AI team performance.
>
>
> - We examined the performance of humans after manipulating explanations and compared it to the performance of a "fixed AI model." In the recidivism prediction task, we found that humans with manipulated explanations achieved an average accuracy of 0.66, which was higher than the fixed AI model's accuracy of 0.62. However, in other tasks, human performance was lower than that of the fixed AI model.
>
> > In addition, the reliance metrics of Fig. 3 are not defined here, and the readers need to search the referenced papers for definitions.
>
> We will revise the manuscript as suggested to include the definition of the reliance metric.
>
>
> [1]  Kerrigan, Gavin, Padhraic Smyth, and Mark Steyvers. "Combining human predictions with model probabilities via confusion matrices and calibration." Advances in Neural Information Processing Systems 34 (2021): 4421-4434.

---

### Official Review · Reviewer_L1hd · 2024-07-12

**Soundness:** 4
**Presentation:** 4
**Contribution:** 3
**Rating:** 6
**Confidence:** 3

**Summary:**

This paper proposes a novel method that manipulates human decision-making by manipulating AI explanations in human-AI interaction scenarios. By utilizing human behavior models and minimizing the cross-entropy function between human and AI agreement with constraint to generating the same AI recommendation outcomes, the authors investigate this method for adversarial and benign purposes. The results show that the manipulation in AI explanations significantly negatively affects human decisions in the four tasks with adversarial purpose and enhances human decision accuracy, over-reliances, and diminishes under-reliances. The paper also discusses how the human perception to the AI explanations varies with the manipulation. Overall, it provides a novel and practical way to intervene in human decision-making in human-AI interactions.

**Strengths:**

- The paper formulates the AI explanation manipulation problems, as optimization problems to minimize human-AI decision disagreement, with constraint to keep the original AI outcomes. This can indeed be applied in both positive or negative ways (in respective to adversarial and benign purposes in the paper). The authors discuss both sides of the methods and provide an implication for society about such methods.

- The paper selects multiple AI explanation baselines and collects human data on multiple tasks, enhancing the robustness of the manipulation method and broadening potential social impacts in various scenarios. The paper also collects human perception, providing an extra view from human participants in subjective feelings.

**Weaknesses:**

- Only particular features are selected in each task. This may weaken the effectiveness of the manipulation. For example, in the census task, the paper selects 'gender' as the dimension of manipulation; while 'degree of education' or 'ages' can possibly be biased and effect human decision-making. Thus, the authors may need to address such representativeness (or probably other dimensions do not exhibit significant effects in manipulations).

**Questions:**

- My question is humans themselves may have a lot of biases, for example, peak and anchoring effects; or in risky decision-making, humans can have probability distortion. This is to say, for humans, the change of 'numerical values' may change their perception of certain things. Probably the manipulations here are not directly related to the importance. So how would the authors know whether the manipulation takes effect on the importance but not simple number perception? (For example, changing from positive to negative, or changing small numbers to large numbers; changing the rank in overall dimensions). This may require a baseline test, whether about the numerical values (in an irrelevant context) or about other measures of importance (e.g., relative rank, or a Likert scale on each dimension separately).

**Limitations:**

- The limitation of this paper as discussed, is only certain features in each task are selected to manipulate. It would reveal more about the societal impact if comprehensive features are considered. Moreover, the consideration of interactions between features is also helpful, though this does require a larger dataset.

---

> ### Author Rebuttal · Authors · 2024-08-05
>
> Thanks for the review! Below, we address your questions.
>
> > Only particular features are selected in each task. This may weaken the effectiveness of the manipulation.
>
> - Thank you for your insightful feedback. Firstly, we’d like to clarify that in Section 5 (Evaluation I, when AI explanations are manipulated for adversarial purposes), the chosen “targeted” feature is only used for defining the goal of manipulation (i.e., whether the human decision makers’ decisions are “fair” or not is defined with respect to the chosen feature); it does not mean that we only manipulate the explanation for that feature. In Section 6 (Evaluation II, when AI explanations are manipulated for benign purposes), no feature is “selected” because the goal of the manipulation is to improve the accuracy of the human decision makers’ final decisions. Again, on each task instance, our manipulation can occur on any subset of the features.
>
> - In addition, for Section 5, to validate the effectiveness of adversarial manipulation of AI explanations, conducting large-scale human experiments is essential; however, such experiments are both costly and resource-intensive. Therefore, for each type of decision making task, we focused on manipulating the most sensitive feature, which we hypothesized would have the most significant societal impact on decision making.
>
> - To ensure the generalizability of our results , we conducted evaluations across four different types of decision making tasks. While we only selected one feature per task to define our fairness goal, we believe that similar manipulative effects could be observed if fairness is defined by other features. This belief is based on our successful manipulation of human behavior for both benign and adversarial purposes, where participants were unable to perceive any differences in the presented manipulated explanations.
>
> > My question is humans themselves may have a lot of biases, for example, peak and anchoring effects; or in risky decision-making, humans can have probability distortion. This is to say, for humans, the change of 'numerical values' may change their perception of certain things.
>
> - Human bias in decision making: Our manipulation is based on a learned human behavior model, which inherently captures the biases present in human decision-making, such as anchoring effects, probability distortion, and other cognitive biases (if they exist). Since these biases are integrated into the behavior model, our manipulation leverages them to nudge human decision makers to make the desired decisions.
>
> -  Human Interpretation of Number: During the tutorial phase of our experiment, we carefully followed established HCI research practices to educate participants on how to interpret AI explanations. We specifically instructed them that the numerical values or bar lengths associated with each feature explanation represent the feature's importance. A positive value indicates a positive contribution to the decision, while a negative value indicates a negative contribution. This framing ensured that participants understood the numbers as a direct reflection of feature importance, rather than as arbitrary values. Thus, in our study, the change of numerical values was directly tied to perceived feature importance as intended.
>
> - Baseline Test Consideration: We appreciate your suggestion regarding a baseline test. To better understand your suggestion, could you please clarify whether you are suggesting that we conduct a study to determine if participants perceive features with larger numerical values as more important, irrespective of context? We are happy and prepared to conduct this test if it would help address any concerns about the study.

---

> > ### Comment · Reviewer_L1hd · 2024-08-11
> >
> > Thanks for the clarification by the authors.
> >
> > Regarding the last question, yes. The authors could possibly consider conducting a baseline test to probe how the selected participants generally perceive the numbers at the scale in the conducted formal experiment. This may help to control individual biases and provide a more accurate prediction.
> >
> > I think the authors well addressed my concerns. But given the constraint of the overall scope, I will keep my current evaluation.

---

### Official Review · Reviewer_f7Sf · 2024-07-13

**Soundness:** 3
**Presentation:** 2
**Contribution:** 3
**Rating:** 6
**Confidence:** 3

**Summary:**

The authors show that by modeling a human decision maker they can manipulate the provided information in ways that reliable influence their decisions towards even non-benign outcomes.

**Strengths:**

# originality

Manipulating Mturk works is a well studied area of research, but this is a unique approach and highlights the limitations of interpretability research using similar techniques.

# quality

The results are statistically significant and done across a range of tasks which strengthens their claims.

# clarity

Most of the plots are readable and I think I was mostly able to understand the results after reading a few time.

# significance

This is an interesting result, presenting an example of "attacking" these score based systems via manipulating the interpretably metrics is an important result for the XAI literature.

**Weaknesses:**

# originality

Manipulating workers on digital platforms is a well studied area.

# quality

Reading the plots (fig 3) some of these results look to be supporting the null-hypothesis (all interventions have the same effect). Having a table giving the numbers and more details on the statistical tests would make the quality of the results much easier to judge. The results also are weak, in part this is due to a small sample size.

# clarity

If found the plots difficult to read either both being too small and the lack of numbers makes analyzing them tricky. Could the authors include the results as a table in the appendix with the error ranges clearly laid out so we don't have to eyeball tiny error bars.

I also had to refer to the appendix to understand the experimental procedure, making it clearer what exactly is happening in the experiments would greatly aid the readability of this paper.

 # significance

Looking at figure B.2 it looks like the manipulation is mostly making the difference much more extreme and noticeable, this suggests these results could simply be due to the differences in the visual presentation between interventions.

**Questions:**

Are the manipulations working possibly just due to the bar plots being being bigger, and numbers being clearer? So no behavioral model needed.

How should I read figure 1? It looks like it's saying the means 95% confidence intervals overlap, but that they are also below 5% to overlap (bias 1a)?

More generally I think this result is weak, if people could tell the models apart this would be more interesting. As it stands I'm not surprised the authors can manipulate the responses, but I'm not convinced their method is close to optimal.

**Limitations:**

This is an area where ethics should be given additional scrutiny, but I do not see anything alarming in the paper, and the authors take the correct tone in the paper.

---

> ### Author Rebuttal · Authors · 2024-08-05
>
> Thanks for the review! Below, we address your questions.
>
> >  Are the manipulations working possibly just due to the bar plots being being bigger, and numbers being clearer? So no behavioral model needed.
>
> - Our behavior model-based manipulation demonstrated that changing the bar length in some task instances—indicating a change in the perceived importance of a feature—indeed influences human decision making. This shift in behavior, captured by our human behavior model, aligns with previous research which showed that overstating model confidence can similarly impact decisions [1,2,3].
>
> - However, our manipulation strategy goes beyond merely increasing the length of bars to make them more salient. In some cases, the manipulation also involves reducing the bar length or even reversing the direction of the bars to downplay or misrepresent a feature's importance.  We observed that in many cases, especially when we manipulate explanations for adversarial purposes and AI models may actually be unfair, the model-based manipulation intentionally retains certain features as protective to mislead human decision making. For example, for the census prediction task, when predicting whether a person's income is below \\$50k, if the person is a female, the manipulated explanation might retain gender as a positive contributor to reaching the \\$50k threshold, whereas for males, gender might be depicted as a negative factor. By doing so,  humans are more likely to make unfair decisions compared to those who got unmanipulated explanations based on our evaluation data.
>
> Therefore, the manipulations based on behavior models are not solely about making bars longer but are tailored to strategically influence human perceptions in line with the model's understanding of decision making behavior.
>
> > How should I read figure 1? It looks like it's saying the means 95% confidence intervals overlap, but that they are also below 5% to overlap (bias 1a)?
>
> In Figure 1(a), the circle (or square or rectangular) represents the mean values, and the error bars represent the 95% confidence intervals. This plot is intended to illustrate how the average False Positive Rate Difference (FPRD) or False Negative Rate Difference (FNRD) of human decisions might be distributed within each of the three treatments (i.e., Manipulated, LIME, SHAP). The statistical significance between a pair of treatments is directly determined by the linear regressions, rather than relying on the visual check of the overlap of confidence intervals. Significance levels are based on the p-values from t-tests on the coefficients derived from these regression models.
>
> > More generally I think this result is weak, if people could tell the models apart this would be more interesting. As it stands I'm not surprised the authors can manipulate the responses, but I'm not convinced their method is close to optimal.
>
> If we understand your comment correctly, you are suggesting that the results would  be more interesting if participants could tell that the explanations had been manipulated. Additionally, it seems you think that the current results might simply be due to the fact that the bars in the explanations were made longer, as mentioned in your earlier point.
>
> - We would like to clarify that our manipulation is not simply about making the bars longer. Instead, as discussed in response to the first question, our approach is based on a human behavior model and is strategically tailored to influence human perceptions in alignment with the model's understanding of humans’ decision-making behavior (i.e., how humans will factor AI recommendations and explanations into their final decisions).
>
> - Regarding whether people can tell if the explanations are manipulated, we find it is also interesting—and more concerning—that human behavior can be changed without participants detecting these manipulations, which indicates significant implications for secure and reliable human-AI collaboration. If participants are unaware of the subtle influences on their decision making, it raises important questions about the transparency and trustworthiness of AI systems, particularly in contexts where security and reliability are paramount.
>
> > Paper clarity
>
> Finally, we will revise the manuscript to enhance the readability of the figures by increasing their size and adding the corresponding labels. Additionally, we will include the linear regression results as tables in the appendix.
>
>
> [1] Vodrahalli, Kailas, Tobias Gerstenberg, and James Y. Zou. "Uncalibrated models can improve human-ai collaboration." Advances in Neural Information Processing Systems 35 (2022): 4004-4016.
>
> [2] Zhang, Yunfeng, Q. Vera Liao, and Rachel KE Bellamy. "Effect of confidence and explanation on accuracy and trust calibration in AI-assisted decision making." Proceedings of the 2020 conference on fairness, accountability, and transparency. 2020.
>
> [3] Rechkemmer, Amy, and Ming Yin. "When confidence meets accuracy: Exploring the effects of multiple performance indicators on trust in machine learning models." Proceedings of the 2022 chi conference on human factors in computing systems. 2022

---

> > ### Comment · Reviewer_f7Sf · 2024-08-09
> >
> > Thank you for the clarification. I've read your response and the other reviews/responses. I still think this work is a marginal accept and maintain my score, but will check back if there is more discussion.

---

### Decision · Program_Chairs · 2024-09-25

**Decision:**

Accept (poster)

**Comment:**

The paper presents a novel approach to manipulating human decision-making through AI-generated explanations, exploring both benign and adversarial uses. The method's originality and robust experimental validation across multiple tasks make it a valuable contribution to the field of explainable AI (XAI).

However, several reviewers raised ethical concerns, particularly regarding participant compensation in the user study and potential biases in datasets. Addressing these issues in the final version would be necessary.

In addition, the paper would benefit from improvements in clarity and comprehensiveness. Reviewers noted that the presentation of results could be enhanced, particularly with more detailed statistical analyses and clearer visualizations. Expanding the scope of experiments to include more complex decision-making scenarios and providing access to the data and code for replication would further validate the findings.